# Results from a cross-specialty consensus on optimal management of patients with chronic kidney disease (CKD): from screening to complications

Mustafa Arici ,[1] Samir Helmy Assaad-Khalil,[2] Marcello Casaccia Bertoluci,[3] Jason Choo,[4] Yau-Jiunn Lee,[5] Magdalena Madero,[6] Guillermo Javier Rosa Diez,[7] Vicente Sánchez Polo,[8] Sungjin Chung,[9] Teerawat Thanachayanont,[10] Carol Pollock[11]

For numbered affiliations see end of article.

**Correspondence to**
Dr Mustafa Arici;
marici@hacettepe.edu.tr

## ABSTRACT

**Background** Chronic kidney disease (CKD) affects around 10% of the global population and has been estimated to affect around 50% of individuals with type 2 diabetes and 50% of those with heart failure. The guideline-recommended approach is to manage with disease-modifying therapies, but real-world data suggest that prescribing rates do not reflect this in practice.

**Objective** To develop a cross-specialty consensus on optimal management of the patient with CKD using a modified Delphi method.

**Design** An international steering group of experts specialising in internal medicine, endocrinology/diabetology, nephrology and primary care medicine developed 42 statements on aspects of CKD management including identification and screening, risk factors, holistic management, guidelines, cross-specialty alignment and education. Consensus was determined by agreement using an online survey.

**Participants** The survey was distributed to cardiologists, nephrologists, endocrinologists and primary care physicians across 11 countries.

**Main outcomes and measures** The threshold for consensus agreement was established a priori by the steering group at 75%. Stopping criteria were defined as a target of 25 responses from each country (N=275), and a 4-week survey period.

**Results** 274 responses were received in December 2022, 25 responses from Argentina, Australia, Brazil, Guatemala, Mexico, Singapore, South Korea, Taiwan, Thailand, Turkey and 24 responses from Egypt. 53 responses were received from cardiologists, 52 from nephrologists, 55 from endocrinologists and 114 from primary care physicians. 37 statements attained very high agreement (≥90%) and 5 attained high agreement (≥75% and <90%). Strong alignment between roles was seen across the statements, and different levels of experience (2–5 years or 5+ years), some variation was observed between countries.

**Conclusions** There is a high degree of consensus regarding aspects of CKD management among healthcare professionals from 11 countries. Based on these strong levels of agreement, the steering group derived 12 key

### STRENGTHS AND LIMITATIONS OF THIS STUDY

⇒ 274 responses received with good representation from experienced physicians across 11 different countries.

⇒ The majority of responses from physician with more than 10 years experience in role, suggesting that the results represent the views of experienced physicians.

⇒ Very high levels of agreement may suggest either that the statements were constructed to achieve agreement (confirmation bias), or that they represent already recognised good practice.

⇒ Lack of representation from low-income or middle-income countries (LMICs), therefore, these findings may be of limited applicability to LMICs and a further Delphi process would be required to capture this perspective.

⇒ Aspects of patient choice and empowerment and consideration of the patient experience (outside of treatment outcomes) have not been discussed.

recommendations focused on diagnosis and management of CKD.

## BACKGROUND

Chronic kidney disease (CKD) is defined as abnormalities of kidney structure or function, which are present for >3 months.[1] This is indicated by a glomerular filtration rate (GFR) below $60\,\mathrm{mL/min/1.73\,m^2}$ or the presence of one or more markers of kidney damage. CKD is classified based on cause, GFR category and albuminuria category.[1] Diabetes is the leading cause of CKD, followed by hypertension, suggesting a close relationship between the cardio-renal-metabolic (CRM) systems.[2] The interplay between CKD, cardiovascular disease (CVD) and metabolic diseases such as type 2 diabetes mellitus (T2DM) is significant and has been termed 'CRM' disease.[3] It has

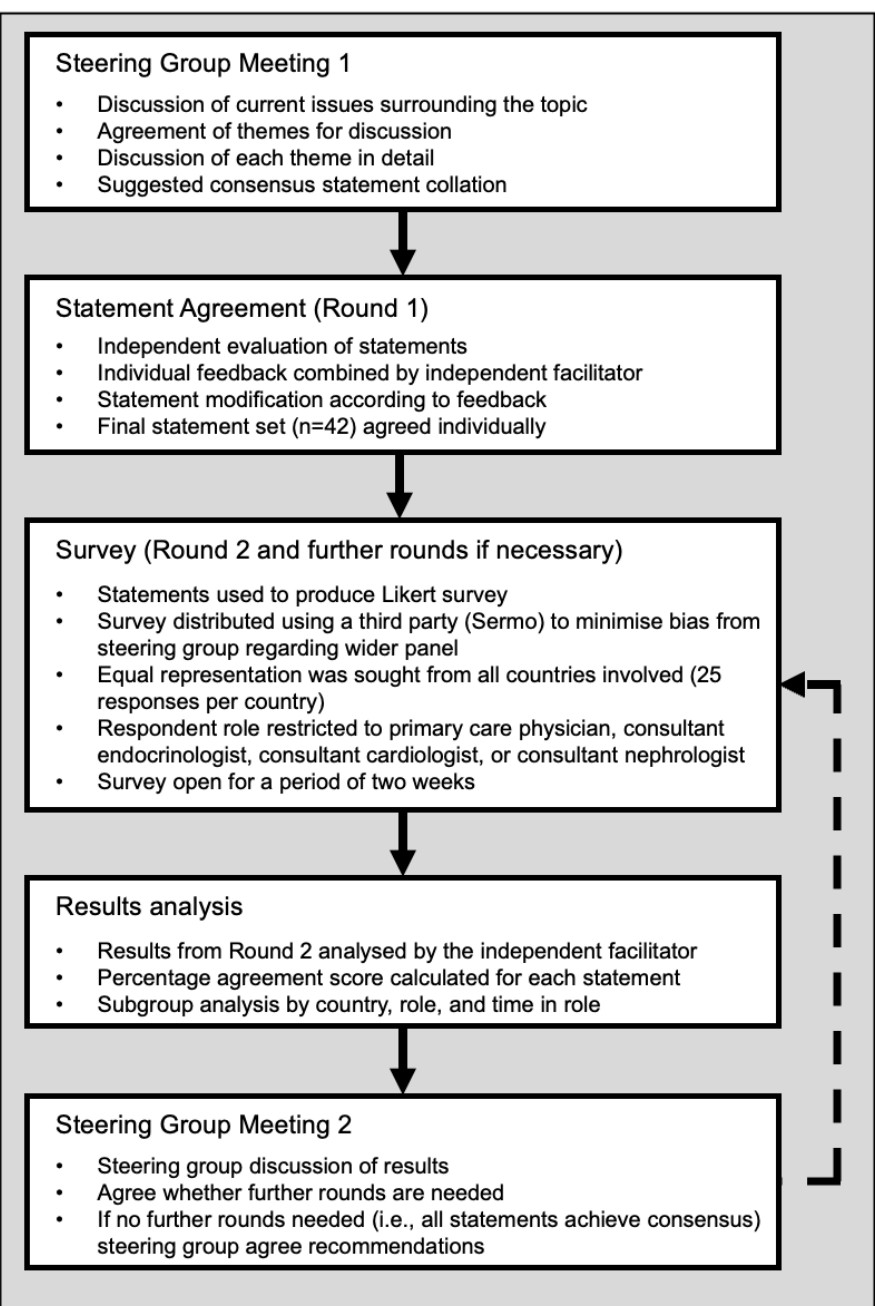

**Figure 1** Study flow diagram.

been suggested that CKD affects around 50% of people with T2DM,[4] around 50% of heart failure (HF) patients,[5] and up to 38% of those with hypertension,[6] suggesting the need for a holistic approach to management of patients with CKD.

Between 1990 and 2017, the prevalence of CKD was estimated to rise to 9.1% of the global population, with associated mortality increasing by 41.5%.[7] This is closely associated with an increase in populations with risk factors such as diabetes, hypertension and pre-diabetes.[2] As a consequence of these factors, 1.2 million people died in 2017 as a consequence of CKD.

Early diagnosis and referral of CKD are key to reducing or avoiding progression to kidney failure, and reducing morbidity and mortality. The absence of symptoms in early stages of CKD requires that clinicians maintain suspicion in all patients, especially those with risk factors.[8] Accurate screening, diagnosis and risk stratification need to be in place to support the aim of 'earliest diagnosis'.

A diagnosis of CKD is determined by laboratory confirmation of proteinuria or haematuria, and/or a reduction in the GFR, for more than 3 months duration.[9] Two key markers of CKD include albuminuria (defined as a urinary albumin-to-creatine ratio (uACR) of >30 mg/g) and reduction in estimated GFR (eGFR), as calculated by the CKD-EPI based on creatinine alone (eGFR$_{cr}$) or on creatinine and cystatin C (eGFR$_{cr-cys}$).[9 10] In practice, these methods may not be used effectively, an analysis of

over 123 000 patient records reported a low frequency of uACR testing in patients with CKD, despite strong epidemiological evidence linking increased albuminuria with disease progression, kidney failure, cardiovascular events and premature mortality.[11]

Besides lifestyle modifications, disease-modifying therapies (DMTs) for CKD including renin–angiotensin–aldosterone system inhibitors (RAASi) and sodium-glucose co-transporter-2 inhibitors (SGLT2i), as recommended by Kidney Disease Improving Global Outcomes (KDIGO) for use in patients with CKD with hypertension or diabetes, and non-steroidal mineralocorticoid antagonists as add-on therapy in patients with T2DM and residual albuminuria.[1 12] Despite the availability of these therapies, analysis of two large US healthcare systems showed that even though nearly two-thirds of the adults with CKD had diabetes, hypertension or pre-diabetes, rates of prescribing RAASi were low. Not only was DMT use below guideline-directed levels, but potentially nephrotoxic agents (such as non-steroidal anti-inflammatory drugs and proton pump inhibitors) were used more commonly than RAASi.[2]

There is clearly some disconnect between guideline recommendations and real-world practice. Guidelines for the management of CKD may not be clear or made known to primary care practitioners (PCPs). The objective of this project is to use a modified Delphi technique to examine the opinions of healthcare professionals (HCPs) towards aspects of CKD management across 11 countries, report these findings and develop practical recommendations for diagnosis and management of CKD.

## METHODS

A steering group of experts (see author list) from 11 different countries convened in September 2022 to discuss current challenges in CKD management. The experts were defined as specialists in nephrology, endocrinology/diabetology, internal medicine and primary care medicine who had achieved an appropriate level of seniority within their field (eg, professors, clinical directors), or had published papers related to the management of CKD/HF/Diabetes, or had been involved in guidelines development.[13–21] The steering group members were selected to provide representation across a range of development indicator levels external to Europe and North America (to avoid replication of previous published work[22]). Steering group members were recruited to represent countries outside of North America and Europe. Central and South America, Southeast Asia, Middle East and Africa were initially targeted. The process of recruitment involved identification of potential group members from each of these regions using desk research, followed by a snowball method until all target regions had at least one representative on the steering group. Although a wide representation was aimed, it is a limitation that there were not enough members from low-income countries.

The Delphi technique used in this study was guided by Guidance on Conducting and REporting DElphi Studies. While guidelines exist, outcomes for CKD vary between countries, a modified Delphi approach was employed to understand where common care processes differ in local use, and how the attitudes of HCPs to elements of CKD management differ between countries. The overall process is outlined in figure 1.

Six themes for focus for statement development were agreed (table 1), these were discussed further, and statements developed by the steering group working collaboratively. The statements were then collated, and the steering group independently rated the statements as either 'accept', 'remove' or 'reword' with suggested changes (as determined by a simple majority) with the potential for further group consultation for any significant differences of opinion on the fundamental principles of any statement. Once finalised, the steering group was agreed the final set of statements for testing. This constituted the initial round of consensus.

The resulting 42 statements were developed into a Likert survey, which was then distributed by a third party (Sermo) in round 2 of the process.

Panel members were recruited based on the following criteria:

► Employed within 1 of the 11 target countries.
► 25 respondents from each country in a broadly 2:1:1:1 ratio of primary care physician, consultant endocrinologist, consultant cardiologist, consultant nephrologist (or local equivalent).

The identity of respondents was not known to the steering group or the independent facilitator. For each statement, respondents were offered a 4-point scale ('strongly disagree', 'tend to disagree', 'tend to agree' and 'strongly agree') to indicate their level of agreement with each statement. The survey also captured country, specialty, length of time in role and average number of patients with CKD managed over 3 months. Stopping criteria for data collection were defined as a target of 25 responses from each country (N=275) and a 4-week survey period.

The target countries were chosen to reflect the steering group demographic, this would enable each steering group member to provide insight into results from their respective country. Panel members were recruited from the following countries: Argentina (UMI), Australia (HI), Brazil (UMI), Egypt (LMI), Guatemala (UMI), Mexico (UMI), Singapore (HI), South Korea (HI), Taiwan (HI), Thailand (UMI) and Turkey (UMI) (HI—high income, LMI—lower-middle income, UMI—upper-middle income, according to World Bank Classification.)

The closing criteria for the study were defined *a priori* in line with best practice principles as: 90% of the final statement set achieving consensus threshold (defined at 75%, a widely accepted threshold).[23] If these criteria were not met, statements would be modified, and the survey reissued as necessary for a maximum of three rounds. Consensus was further categorised as 'high' at

**Table 1** Agreement to each statement, divided by topic

| No: | Statement | Overall |
|---|---|---|
| **Earlier identification and screening of CKD** | | |
| 1. | Late diagnosis is a lost opportunity for early management of CKD which can slow disease progression. | 93% |
| 2. | National screening and diagnostic programmes for CKD are essential to identifying patients at earlier stages of CKD. | 96% |
| 3. | A simple and practical definition of 'high-risk' should be established to support cost-effective screening. | 97% |
| 4. | Early screening for CKD in high-risk groups is cost-effective for the health system (where resources are in place to support intervention). | 97% |
| 5. | Primary care physicians and specialists (cardiologists, endocrinologists, nephrologists, diabetologists, etc) should routinely screen for CKD in patients with risk factors. | 97% |
| 6. | Patients with risk factors (eg, elderly, diabetes, hypertension, dyslipidaemia, obesity, family history of CKD) should be screened at least annually for CKD. | 97% |
| 7. | Patients with CKD should be screened for heart failure (including those with heart failure with preserved ejection fraction). | 92% |
| 8. | GFR estimated by the CKD-EPI Creatinine Equation (2021) and albuminuria (using albumin-to-creatinine ratio) should be the screening method of choice for CKD. | 95% |
| 9. | Confirmatory testing of abnormalities of kidney function or structure at 3 months is required to establish a CKD diagnosis. | 92% |
| **Risk factors for CKD in cardio-renal-metabolic patients** | | |
| 10. | A holistic view of cardio-renal-metabolic disease states is critical to provide integrated patient-centred care to individuals with CKD. | 99% |
| 11. | CKD progressively increases the risk of CVD, including heart failure and overall mortality. | 99% |
| 12. | Frequency of hyperkalaemia increases as CKD progresses. | 97% |
| 13. | AKI is a risk factor for CKD and chronic heart failure. | 93% |
| 14. | Infection, including COVID-19, can cause AKI and is therefore a risk factor for CKD. | 85% |
| 15. | Nephrotoxic medications should be modified, and risk/benefit profiles should be assessed in patients with, or at risk of CKD. | 98% |
| **Holistic management of CKD in cardio-renal-metabolic patients** | | |
| 16. | Lifestyle factors such as diet, smoking, exercise must be optimised for patients with CKD. | 100% |
| 17. | Disease-modifying therapies for CKD (eg, SGLT2i, RAASi, MRA) should be used where indicated to manage progression of CKD. | 99% |
| 18. | SGLT2i and RAASi have a complementary cardiorenal protective action. | 98% |
| 19. | Early use of SGLT2i could prevent the development and progression of patients with CKD with type 2 diabetes and heart failure. | 96% |
| 20. | Early use of SGLT2i can slow progression of CKD in patients without diabetes. | 91% |
| 21. | A small decrease of eGFR (~10%) may be expected on initiation of CKD disease-modifying therapies. | 93% |
| 22. | De-escalation or discontinuation of RAASi therapy is associated with worse cardiovascular and renal outcomes in the patient with CKD. | 90% |
| 23. | Disease-modifying therapies for CKD should only be stopped as a last resort. | 82% |
| 24. | Novel K binders (ie, patiromer and sodium zirconium cyclosilicate) are an option to manage hyperkalaemia and prevent de-escalation or downtitration of RAASi. | 91% |
| 25. | Action to manage hyperkalaemia in the patient with CKD should be taken when serum potassium level reaches 5.0 mmol/L. | 75% |
| 26. | Action to manage hyperkalaemia in the patient with CKD should be taken when serum potassium level reaches 5.5 mmol/L. | 93% |
| 27. | Treatment targets such as blood pressure, blood glucose, lipid levels should be maintained for both kidney and cardiac outcomes in patients with CKD. | 98% |
| **Guidelines** | | |
| 28. | Implementation of evidence-based CKD guidelines is suboptimal and should be improved. | 77% |
| 29. | Cardiology, nephrology and endocrinology guidelines for CKD should be aligned. | 98% |

Continued

**Table 1** Continued

| No: | Statement | Overall |
|---|---|---|
| 30. | Primary care physicians have an important role in implementing guidelines. | 95% |
| 31. | Guidelines should be practical with an executive summary/checklist to assist implementation by non-specialist HCPs. | 97% |
| 32. | Guidelines should include clear criteria for when and how to refer to other specialists/MDT. | 98% |
| 33. | Guidelines should reflect differences in population phenotypes and be locally adapted where needed. | 94% |
| 34. | Patient perspective is crucial in developing guidelines. | 88% |
| 35. | Investment is needed to shift the approach to primary prevention rather than end-stage kidney disease management. | 96% |
| **Cross-specialty alignment (cardiology, nephrology, endocrinology, primary care and policy-makers)** | | |
| 36. | Patients with CKD should be managed by an MDT where possible with an agreed management plan. | 99% |
| 37. | The CKD MDT should include the primary care physician to improve early intervention and decision making. | 97% |
| 38. | The care pathway for CKD should be designed to minimise the impact of organisational barriers. | 98% |
| 39. | Clinicians, professional associations, academic institutions and patient representative organisations need to engage with policy-makers to ensure appropriate plans and funding are in place to deliver optimal CKD care. | 98% |
| **Education of clinicians and patients** | | |
| 40. | Up-to-date HCP education on the management of patients with CKD and associated guidelines is needed. | 99% |
| 41. | Structured education of primary care physicians on screening and early detection improves patient outcomes. | 100% |
| 42. | Therapeutic patient education is key for them to understand the consequences of CKD and how to manage it through lifestyle modification and appropriate use of therapies | 99% |

AKI, Acute Kidney Injury; CKD, chronic kidney disease; CKD-EPI, Chronic Kidney Disease Epidemiology Collaboration; CVD, cardiovascular disease; eGFR, estimated GFR; GFR, glomerular filtration rate; HCPs, healthcare professionals; MDT, multidisciplinary team; MRA, mineralocorticoid antagonist; RAASi, renin–angiotensin–aldosterone system inhibitors; SGLT2i, sodium-glucose co-transporter-2 inhibitors.

≥75% and 'very high' at ≥90%. Completed surveys were analysed using Microsoft Excel software. The responses were aggregated to provide an overall agreement level (ie, the number of responded expressing agreement as a percentage of the overall number of responses for each statement).

### Patient and public involvement

None, the stated objective was to examine the opinions of HCPs towards aspects of CKD management across 11 countries.

### RESULTS

Of the 45 initial statements created by the steering group, 32 were retained, 8 were modified, 5 were removed and 3 new statements were added. The final set was then sent out to the group via email for acceptance prior to progressing to round 2.

In round 2, completed surveys from a panel of 274 were analysed to define the total level of agreement with each of the 42 statements. Respondents were either PCPs or consultants in cardiology, nephrology and endocrinology (including diabetology) (figure 2). Most responders had more than 10 years experience in role (159/274) and only 13% (n=35) had less then 5 years in role. When asked to estimate the number of patients seen in a 3-month period, 177 responders (65%) stated that they saw more than 50 patients (either as inpatients or outpatients). This suggests the respondent cohort was sufficiently experienced to provide insight into aspects of CKD.

Consensus was achieved for all statements, with very high agreement (>90%) in 37 (88%) statements, and high agreement (≥75 and ≤89%) in 5 (12%) statements. Overall consensus agreement by statement is shown in figure 3 and table 1, detailed results showing percentages

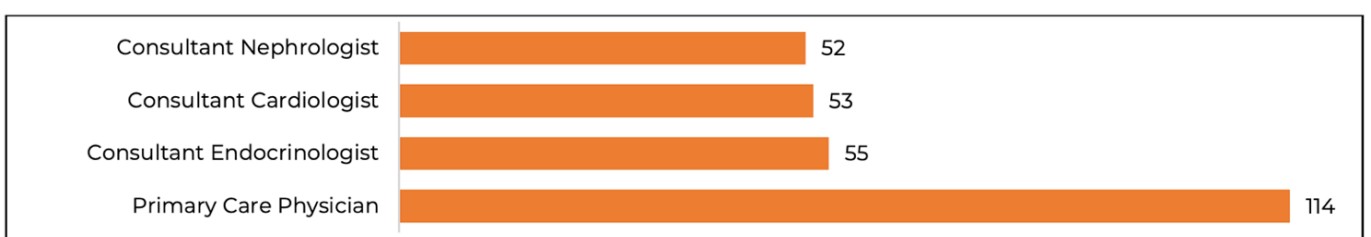

**Figure 2** Respondents by role.

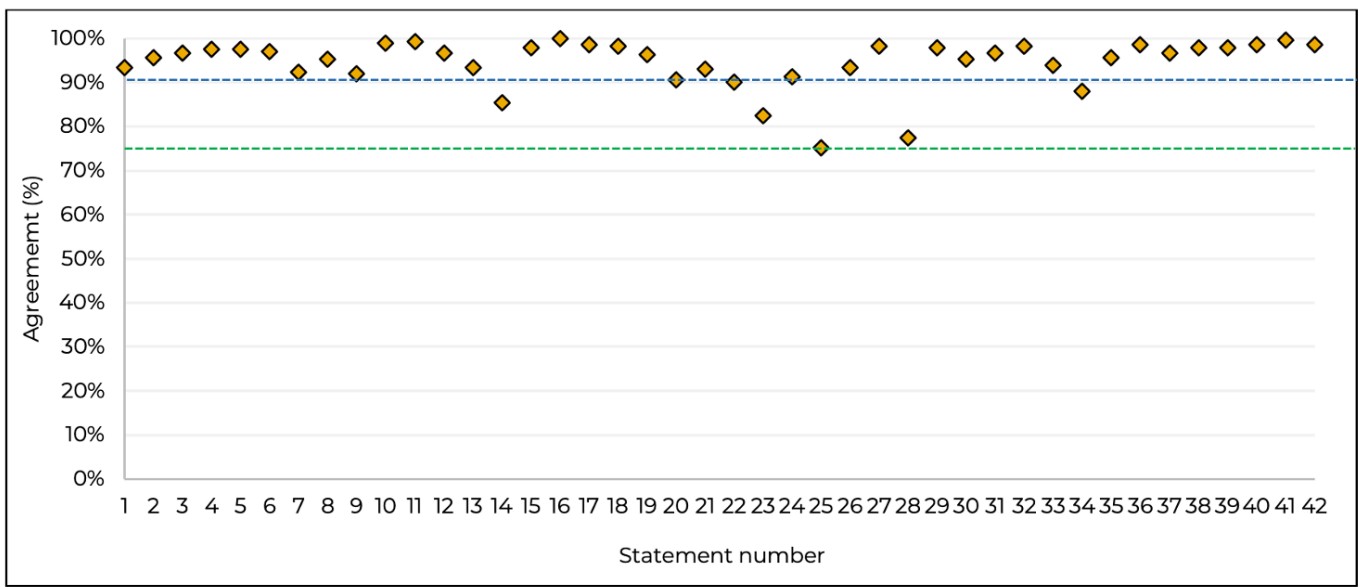

**Figure 3** Overall consensus agreement levels by statement.[a] [a]The green horizontal line represents the 75% threshold for consensus agreement and the blue line indicates the threshold for very high consensus (90%).

of agreement at each level can be found in online supplemental table 1 and online supplemental figure 1.

As none of the statements failed to reach the predetermined threshold of 75%, only one round of survey was required. The results of the survey represent current opinions of the respondents and are not intended to contradict the established evidence base. Anonymised round 2 results data are available in online supplemental table 2.

## DISCUSSION

Note, in the discussion below, 'S' is used to denote 'statement'

### Earlier identification & screening of CKD (S1-9)

Very high agreement (>90%) was observed for all statements in this topic, underscoring the key principle that early diagnosis of CKD is key to implementing strategies to slow disease progression. Responses suggest a strong support for the need for national kidney health screening and diagnostic programmes. Universal screening of the general population has been found not to be cost-effective,[24] but systematic review has shown screening to be cost-effective in patients with hypertension or diabetes.[25] Indeed, a KDIGO Controversies Conference concluded that targeted groups, such as those with hypertension, diabetes or CVD should be screened for CKD, and that an individualised approach should be taken to screen others based on a range of factors.[26]

Respondents agree that patients with risk factors should be screened for CKD at least annually by using eGFR and uACR where available, but identification of CKD is also challenging where awareness among healthcare staff and health literacy in the general population is poor, it is, therefore, recommended that national initiatives

to improve these issues should support any screening programme.

A minimal-resource prescreening tool has been developed and globally validated for CKD in people with T2DM. This demonstrates that age, gender, body mass index, duration of diabetes and blood pressure information can be used to identify those at an increased risk of CKD.[27] This model does not require sophisticated diagnostics and can be used to guide cost-effective screening for CKD where resources are limited.

### Risk factors for CKD in CRM patients (S10–15)

All statements in this topic achieved >90% consensus apart from statement 14 (85%). This is interesting and could be due to the wording of the statement and the specific use of COVID-19 as an example. Reference to COVID-19 was included in the statement as it was considered relevant by the steering group at the time of statement generation and evidence was beginning to emerge of potential kidney injury associated with COVID-19. Analysis by country found that Taiwan agreement was at 68% for this statement, noticeably lower than for other countries. Although 93% of respondents agree that acute kidney injury (AKI) is a risk factor for CKD and HF, it is interesting that 7% do not agree given the evidence base.[28 29]

Given the established interplay between the cardiac, renal and endocrine systems, it is heartening to see that 99% of respondents agree that a holistic approach is needed to provide personalised care for individuals with CKD.

Where practical, models of care should be developed to deliver integrated multidisciplinary care to patients with CRM disease, as described by the CardioMetabolic Care Alliance, to use comprehensive, patient-centred,

team-based approaches for aggressive secondary prevention.[30] Implementing combined clinics to deliver medical care for patients with kidney disease and diabetes or CVD may reduce outpatient healthcare costs without compromising health outcomes.[31] Combined multidisciplinary clinics for diabetes, CKD and CVD are also associated with a slower decline in GFR than usual care, and a significant reduction in the risk for all-cause mortality.[32] However, global variation exists regarding resource availability and access to specialist care,[33] and the use of multidisciplinary teams (MDTs) is not universal. In these regions, HCPs are encouraged to develop local contacts/networks to allow for interdisciplinary discussions of CKD cases where needed.

### Holistic management of CKD in CRM patients (S16–27)

The broad objective of this consensus is to promote the earlier detection and management of CKD, and this principle applies regardless of resource limitations. As CKD-associated healthcare costs increase with disease progression,[34] the economic argument supports this approach, and in patients with limited access to specialist care (including DMT), the role of patient education is paramount in improving health literacy and promoting lifestyle changes to reduce CKD risk. Coupled with this, the health system should look at how interventions such as DMTs can be used to slow progression of CKD and the need for dialysis (and associated costs), cost savings from slower progression to dialysis dependence might be reinvested in detection and diagnosis/national screening of high-risk patient groups.

Very high agreement was observed for 10 of the statements in this topic, with 2 statements at lower levels of agreement (S23 and S25, 82% and 75%, respectively). Response to statement 23 is interesting, given the current debate regarding the benefits of stopping RAASi in advanced CKD. Results from a 52-week open-label UK study in patients with CKD stages 4–5 concluded 'The STOP-ACEi trial did not find any benefit by stopping RAASi in advanced CKD', and that 'stopping RAASi in advanced CKD at an arbitrary GFR threshold is not the optimal approach'. Large observational studies have confirmed the cardioprotective benefits of RAASi in advanced CKD (including patients with concomitant T2DM).[35–37] Linked to this is statement 21, respondents agree that a small decrease in eGFR is to be expected on initiation of DMT for CKD. During initiation and uptitration of RAASi treatment, a decline in kidney function of up to 30% within 4 weeks can be acceptable[1 38] and it is important to avoid a knee-jerk response and reduce or stop DMT, and consultation with a nephrologist or MDT is recommended.

Hyperkalaemia (HK) is potentially life-threatening; it may be acute or chronic and individuals with CKD are at an increased risk, which increases with the later stages of CKD.[39] HK treatment is often stratified by serum potassium level, with borderline levels warranting modification of dietary potassium intake, followed by pharmaceutical management for larger or more sustained increases.[40] Evidence to support recommending low potassium diets to patients with advanced CKD or ESRD is weak. Observational studies report weak associations between dietary potassium intake and potassium concentration and this approach may deprive patients of the beneficial cardiovascular effects associated with potassium-rich diets.[41]

When HK occurs, RAASi dose reduction or discontinuation are the most common used therapeutic options, but this approach is associated with worse cardiorenal outcomes and increased mortality.[42–44] Once stopped, RAASi treatment is rarely reinitiated.[45] In cases of mild-to-moderate HK, DMT should be maintained where possible,[1 14] and an option for achieving this is using a potassium-lowering therapy such as patiromer or sodium zirconium cyclosilicate (S24, 91%).

While the threshold for intervention in HK may vary by country and even individual physician, we can conclude that action may be considered when $K^+$ concentration reaches 5.0 mmol/L (S25, 75%) and certainly when at 5.5 mmol/L (S26, 93%) but with a degree of personalisation (in consultation with a nephrologist). Statements 25 and 26 were included to try and understand where the weight of opinion fell regarding interventions to manage HK, both statements achieved consensus, but the strongest agreement was for statement 26. On reflection, these statements could have been worded better to understand what would constitute appropriate action in these circumstances.

Respondents very strongly agree that SGLT2i and RAASi therapies have a complementary cardiorenal protective action (S18, 98%) and that in patients with T2DM and HF, early SGLT2i use can prevent the development and progression of CKD (S19, 96%), in fact, SGLT2i may slow progression of CKD in patients without T2DM (S20, 91%). It is, therefore, recommended that SGLT2i and RAASi therapies are initiated as early as possible to both delay CKD progression and to therapeutically address the full CRM continuum—including T2DM and HF.

The interplay between CKD, diabetes and CVD has been discussed, but CKD is associated with a number of comorbidities. A prospective UK cohort study of 1741 people with CKD stage 3 found that isolated CKD (without comorbidities) was present in only 4% of patients, and that common comorbidities include 'painful condition' (30%), anaemia (24%), thyroid disorders (12%), cerebrovascular disease (12%) and respiratory conditions (10%). These comorbidities should be identified and managed as part of a multidisciplinary approach.[46]

### Guidelines (S28–35)

While all statements in this topic achieved consensus threshold, the steering group were surprised that 23% of respondents did not agree that implementation of CKD guidelines is suboptimal. The lowest agreement levels were observed in South Korea (68%) and Guatemala (72%). This could be either genuine strength in implementation of guidelines in these countries (which should

be investigated and replicated elsewhere), or the need for continuing professional development even among experienced physicians.

Due to the vital role of primary care must play in identifying and referring suspected patients with CKD, respondents agree that clear guidelines for referral of patients are considered essential (S32, 98%). PCPs should be provided with clear criteria for referral and followed through with continuing education from an appropriate specialist (ideally a nephrologist). The focus on prevention can only be achieved through a whole-system approach with appropriate investment in awareness, screening and diagnosis programmes (S35, 96%).

### Cross-specialty alignment (Cardiology, Nephrology, Endocrinology, Primary Care & Policy Makers) (S35 - 39)

Very high levels of agreement suggest strong support for multidisciplinary involvement in CKD management, and that PCPs should form part of the MDT. Given the prominent status of CKD as an emerging global health threat,[47] there is a need to ensure that plans and funding are in place to enable optimal care. Stakeholders should engage with both local and national policy-makers to ensure that CKD is appropriately prioritised.

### Education of clinicians and patients (S40–42)

Most responders agree on the need for up-to-date education for all HCP roles involved in CKD management, and that PCPs require structured education. Education of patients is a low-cost intervention, to encourage kidney healthy lifestyles and improve awareness of CKD among high-risk populations.

### Strengths and limitations

274 responses received with good representation from experienced physicians across 11 different countries is a strong basis for consensus, the study was designed to ensure that each country was represented by an equal number of responders. The majority of responders reported more than 10 years experience in role (159/274), suggesting that the results represent the views of experienced physicians.

This study had some key limitations. Very high levels of agreement were achieved across the statements, this which may suggest that the statements were constructed as to achieve agreement (confirmation bias), or that perhaps they represent established good practice, a possible improvement to the process might be to provide survey responders (or a subset) with the opportunity to add or amend the statements prior to the full survey. It must also be acknowledged that associations with pharmaceutical companies may have led to unconscious bias the development of the consensus statements, but the study was designed to minimise the impact of this bias on results: the identity of steering group members was not disclosed to survey responders, and the identity of responders was not known to either the facilitator or the steering group.

The lack of representation on the steering group (and subsequently the wider panel) of members from low-income and middle-income countries (LMICs) was a significant limitation. While the survey covered 11 countries of varying economic status, the range was from lower-middle income to high, a clear gap exists, namely those countries classified as 'low income' (eg, sub-Saharan Africa). Consequently, the findings of this study may not be generalisable to LMICs. In order to make such recommendations, a further Delphi process would be required specifically to engage with this demographic, this would provide insight into the specific challenges and considerations and allow for comparison with the current dataset.

Aspects of patient choice and empowerment and consideration of the patient experience (outside of treatment outcomes) have not been discussed, this can be considered a limitation as the patient perspective may have significant bearing on the practicability and implementation of CKD management.

### Recommendations

1. Early screening for CKD in high-risk groups is cost-effective for the health system (where resources are in place to support intervention).
2. GFR (estimated by the CKD-EPI Creatinine Equation) and uACR (using albumin-to creatinine ratio) should be the screening method of choice for CKD.
3. The CKD MDT should include the primary care physician to improve early intervention and decision-making.
4. Intervene early in patients with CKD with proven therapies (ie, SGLT2i, RAASi, DMTs) to delay/prevent progression to kidney failure.
5. RAASi therapies should be optimised in all patients with CKD and HF.
6. Patients with CKD should have their HK managed appropriately when serum potassium level rises above >5.0 mmol/L.
7. Chronic HK should be treated with a potassium binder (eg, patorimer, sodium zirconium cyclosilicate) to allow for the maintenance of DMTs.
8. Guidelines should be practical with an executive summary/checklist to assist implementation by non-specialist HCPs.
9. Guidelines should include clear criteria for when and how to refer to other specialists/MDT.
10. Nephrologists should work together to deliver consistent and clear education regarding CKD management.
11. Clinicians, professional associations, academic institutions and patient representative organisations need to engage with policy makers to ensure appropriate plans and funding are in place to deliver optimal CKD care.
12. National kidney health programmes should be implemented to drive improvements to screening and diagnosis.

## CONCLUSION

The steering group was able to form a set of recommendations specific to the 11 participant countries ranging from lower-middle income to high income and relevant to other countries with a similar demographic. Implementation of these recommendations has the potential to improve detection of CKD at an earlier stage in patients with risk factors. Earlier diagnosis provides the opportunity for early intervention with DMTs that can slow or halt the progression of CKD, in addition to reducing associated morbidity and mortality.

### Author affiliations
[1]Division of Nephrology, Department of Internal Medicine, Faculty of Medicine, Hacettepe Universitesi, Ankara, Türkiye
[2]Unit of Diabetes, Lipidology & Metabolism, Alexandria University Faculty of Medicine, Alexandria, Egypt
[3]Endocrinology Division, Hospital de Clínicas de Porto Alegre, Universidade Federal do Rio Grande do Sul Instituto de Biociencias, Porto Alegre, Brazil
[4]Department of Renal Medicine, Singapore General Hospital, Singapore
[5]Lee's Clinic, Taiwan, China
[6]Division of Nephrology, Department of Medicine, National Heart Institute of Mexico, Mexico City, Mexico
[7]Servicio de Nefrología, Hospital Italiano de Buenos Aires, Buenos Aires, Argentina
[8]Servicio de Nefrología y Trasplante Renal, Instituto Guatemalteco de Seguridad Social, Guatemala City, Guatemala
[9]Division of Nephrology, The Catholic University of Korea College of Medicine, Seoul, Korea (the Republic of)
[10]Nephrology, Bhumirajanagarindra Kidney Institute, Bangkok, Thailand
[11]Kolling Institute of Medical Research, Sydney Medical School, University of Sydney, Sydney, New South Wales, Australia

**Acknowledgements** The authors would like to thank Ian Walker and Dal Singh of Triducive Partners Limited for their assistance in survey distribution, analysis of result, writing the initial draft manuscript and reviewing the final draft.

**Contributors** The authors confirm this manuscript is submitted for publication to BMJ Open as an original article. Neither the entire paper nor any of its contents is currently being submitted or has been accepted by any other journal. We also confirm that the authors meet the criteria for authorship, namely (1) substantial contributions to conception or design of the work, or the acquisition, analysis or interpretation of data for the work; (2) drafting of the work or revising it critically for important intellectual content; (3) final approval of the version to be published and (4) agreement to be accountable for all aspects of the work. MA, SHA-K, MCB, JC, Y-JL, MM, GJRD, VSP, SC, TT and CP agreed the design of the study, formulated and reviewed the statement set. Results of the study were discussed by all contributors as part of a formal steering group meeting where commentary was agreed for development into the initial draft manuscript. MA, SHA-K, MCB, JC, Y-JL, MM, GJRD, VSP, SC, TT and CP took an equal role in reviewing the initial manuscript draft and providing comments, and all approved the final draft. MA is the guarantor for the work and/or conduct of the study, had access to the data and controlled the decision to publish.

**Funding** The study was initiated and funded by AstraZeneca. All authors received funding from AstraZeneca while undertaking this study. AstraZeneca commissioned Triducive Partners to facilitate the project and analyse the responses to the consensus statements in line with the Delphi methodology. After engaging Triducive Partners, AstraZeneca made no contribution to the design and development of the study outside of payment of honoraria. AstraZeneca took no part in the writing, revision or editing of the manuscript except to check that the manuscript contained no promotion of specific medicines and that all recommendations were appropriate to drug label.

**Competing interests** MA has received payment honoraria for lectures, presentations, speaker bureaus, publication writing or educational events from: Amgen, AstraZeneca, Bayer, Boehringer Ingelheim, Menarini, MSD, Novo Nordisk, Sandoz and Sanofi. MA has also participated in a Data Safety Monitoring Board or Advisory Boards for: AstraZeneca, Bayer, Boehringer Ingelheim and Novo Nordisk.
MCB has worked as speaker for: Aché, AstraZeneca, Bayer, Boehringer-Ingelheim, Novo-Nordisk, and Servier. JC has consulted for, or been on advisory boards for: AstraZeneca, ATC, Bayer, GSK, Nitto Denko and Pfizer. JC has been a speaker for: Abbott, AstraZeneca, Bayer and Boehringer Ingelheim. JC has been on Steering Committees or Data Safety Monitoring Boards for: AstraZeneca, Boehringer Ingelheim and Novartis, and has received scientific grant funding from AstraZeneca, ATC, and Nitto Denko. Y-JL has been a speaker and Advisory Board member for: AstraZeneca Taiwan. MM was the International Society for Nephrology Deputy Chair for Latin America and Caribbean (2019–2023) and is currently the on the council for the International Society for Peritoneal Disease (2022–2024). MM has been a consultant for: AstraZeneca, Bayer, and Boehringer Ingelheim, and provided expert testimony for: AstraZeneca, Bayer and Boehringer Ingelheim. MM has received grants from or has pending grants with: AstraZeneca*, Bayer*, Boehringer, Renal Research Institute* and Tricida*. MM has received travel grants from: AstraZeneca and Medcom. GJRD has been a speaker or Advisory Board member for Astra Zeneca, Boehringer Ingelheim, Bayer, Fresenius and Medtronic. SC has received honoraria as a speaker or advisory board member from AstraZeneca, Boehringer Ingelheim, Daewon, Daewoong, Eli Lilly, Korea Pharma, Myoung Poom Medical, Otsuka, Sanofi-Aventis and Yuhan. CP has been a speaker or participated in advisory boards for Astra Zeneca, Boehringer Ingelheim, Eli Lilly, GlaxoSmithKline, Otsuka, Vifor-CSL. SHA-K, VSP and TT declare no conflict of interest in regard to this study.

**Patient and public involvement** Patients and/or the public were not involved in the design, or conduct, or reporting, or dissemination plans of this research.

**Patient consent for publication** Not applicable.

**Ethics approval** As this study only sought the anonymous opinions of physicians and no patient-specific data were captured, ethical approval was not sought. Respondents provided consent to have their responses included in this study through their inclusion to the panel held by Sermo.

**Provenance and peer review** Not commissioned; externally peer reviewed.

**Data availability statement** All data relevant to the study are included in the article or uploaded as online supplemental information.

**ORCID iD**
Mustafa Arici http://orcid.org/0000-0002-4055-7896

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
