## [Reviewer comments · BMJ Open]

ARTICLE DETAILS

TITLE (PROVISIONAL)	Results from a cross-specialty consensus on optimal management of the chronic kidney disease (CKD) patient: from screening to complications
AUTHORS	Arici, Mustafa; Assaad – Khalil, Samir; Bertoluci, Marcello; Choo, Jason; Lee, Yau-Jiunn; Madero, Magdalena; Rosa Diez, Guillermo; Sánchez Polo, Vicente; Chung, Sungjin; Thanachayanont, Teerawat; Pollock, Carol

VERSION 1 – REVIEW

REVIEWER	Luyckx, Valerie Institute of Biomedical Ethics and History of Medicine
REVIEW RETURNED	02-Nov-2023

GENERAL COMMENTS	Arici et al. conducted an 11 country survey of clinicians who manage patients with cardio-renal metabolic syndrome on what they consider important in terms of early diagnosis and management of patients with CKD. This work has significant value in terms of reaching consensus guidelines for management of CKD especially in primary care. The manuscript is clear and well written, the authors are experts in the field. It is disappointing however to see no co-investigators from sub-Saharan Africa, where health systems are the least resourced and where CKD prevalence is arguably highest, disease modifying therapy may in fact have the most life-saving potential given the limitations in access to dialysis/transplant, and access to these therapies is poor. This should be acknowledged and seems a missed opportunity. Could more input be added here from low income settings? The declarations of interest of many the authors of this study having associations with pharma are listed. Given the very high degree of consensus observed here, a potential limitation could be that given that many authors have associations with relevant pharma companies, this could have been a factor which played into the statement development itself (unconsciously). This should be added clearly to the limitations. The CKD statements themselves are not surprising and represent an optimal approach to management. The value here is the collations of the statements together, but most of the statements are not novel. There are some areas where more emphasis would improve the relevance of this manuscript:
---

	1. Many of the suggestions are quite high-resource setting focused. Some discussion of the feasibility of implementation of the recommendations across the country income spectrum and prohibitive medications costs/unavailability should be included in the discussion, for example referencing the Global Kidney Health Atlas report on lack of availability of basic diagnostics at primary care in low resource settings. MDTs are certainly not available world-wide, please include comments about this as well. 2. There is no discussion of coverage for CKD care - the sustainability of the suggested strategies should be discussed, especially thinking about lower resource settings. How should the best be covered financially, integrated into the health system? Quality monitored/assured? 3. More emphasis should be placed on earlier identification more consideration will need to be given to identifying patients with "non-traditional" risk factors, especially in lower resource settings, should be highlighted more strongly. Who is high risk where? 4. The methods needs to be clearer, how were respondents identified/reached? Could this have led to some bias? 5. Why is COVID specifically included in the AKI statement 14? 6. Patient empowerment, health literacy, possibly even patient navigators were not represented adequately in the statements/discussion, this should be included as a limitation 7. A final call for affordability and availability of the recommended medications (i.e. reducing prices) everywhere should be made, despite the study being funded by pharma. Lines 27/28 on page 8 seem incomplete sentences?
--	--

REVIEWER	Kyte, Derek University of Worcester DK reports previous grants from Macmillan Cancer Support, Innovate UK, the NIHR, NIHR Birmingham Biomedical Research Centre, and NIHR SRMRC at the University of Birmingham and University Hospitals Birmingham NHS Foundation Trust, and previous personal fees from Merck.
REVIEW RETURNED	10-Nov-2023

GENERAL COMMENTS	There is a need to define the KDIGO acronym in full on page 7 (line 11) as I believe this is the first time it appears in the main manuscript. The authors should consider reporting the study according to established guidelines - e.g. CREDES https://www.equator-network.org/reporting-guidelines/credes/ - which should be highlighted at the start of the methods section. For completeness (and transparency) ideally, there should be more demographic details regarding the steering group membership and any conflict of interests.
---

	Presenting the theme titles in isolation on pg.8 somewhat hinders interpretation. Some further explanation of each theme is required to give context for later sections. It is a little unclear how the steering group rating of statements, led to the final statement set used for the initial round of consensus. For example, what proportion of 'accept' ratings was required to include a particular statement in the final statement set? The stopping criteria should be justified. Were the criteria selected according to an evidence-based approach? When determining whether a particular statement met the consensus agreement threshold, presumably both the 'tend to agree' and 'strongly agree' responses were aggregated into a single agreement metric? This approach should be outlined in the methods. I note that the authors state 'surveys were collated... in line with Delphi methodology...', but I feel more information is needed here. I feel that many might take issue with the following statement: "Patients and the public were not involved in this research as it is primarily concerned with healthcare policy and practice." Reasoning that patients' opinions and preferences should play a more important role in healthcare policy and practice. With regards the current study, however, arguably the justification for the non-inclusion of patients and the public should be linked more to the stated research objective on pg.6. I feel it is important to breakdown the agreement data in Table 1. It should be clear for each statement what proportion of respondents selected the 'strongly agree' versus the 'tend to agree' option. This allows the reader to explore the differing nature of responses for statements which scored a similar overall level of agreement. For example, both item 9 and item 24 scored similarly for their overall agreement, but the proportions of those respondents strongly agreeing with each statement was quite different (Item 24 = SA 27%, TTA 65%; Item 9 = SA 48%, TTA 44%). In the discussion, some statements required slightly more explanation/qualification as they appeared to extend beyond the scope of the study findings. For example: "Earlier identification & screening of CKD (S1-9) There was very high agreement (>90%) with all statements in this topic, and the key principles that early diagnosis of CKD is key to implementing strategies to slow disease progression. To this end, national kidney health screening and diagnostic programmes should be implemented." It is the last statement here that is somewhat problematic. I feel it might be more accurate to state that the views of the Delphi respondents suggested there was strong consensus supporting the need for national kidney health screening and diagnostic programmes. Adjusting the language in this way serves to keep the recommendations grounded in the study data/findings. I note that this was done at times during the discussion - e.g. pg.15 'Very high levels of agreement suggest strong support for
--	---

	multidisciplinary involvement in CKD management...’ – but, perhaps needed to be a little more consistent throughout. I note the strengths and (one) limitation were presented in bullet point format rather than discussed. The CREDES Delphi reporting guidelines suggest: “Discussion of limitations. Reporting should include a critical reflection of potential limitations and their impact on the resulting guidance.” The authors should consider addressing this in the final manuscript.
--	---

VERSION 1 – AUTHOR RESPONSE

Reviewer: 1

Comments to the Author:

1. Arici et al. conducted an 11 country survey of clinicians who manage patients with cardio-renal metabolic syndrome on what they consider important in terms of early diagnosis and management of patients with CKD. This work has significant value in terms of reaching consensus guidelines for management of CKD especially in primary care. The manuscript is clear and well written, the authors are experts in the field.

Thank you for this feedback

2. It is disappointing however to see no co-investigators from sub-Saharan Africa, where health systems are the least resourced and where CKD prevalence is arguably highest, disease modifying therapy may in fact have the most life-saving potential given the limitations in access to dialysis/transplant, and access to these therapies is poor. This should be acknowledged and seems a missed opportunity. Could more input be added here from low income settings?

Whilst we are not able to provide more input from low-income settings at this point, we agree that this is a significant oversight in the manuscript and have taken steps (as per the points below) to clearly state this limitation and also to provide some more focus in the narrative towards lower income/LMICs which we hope is satisfactory.

3. The declarations of interest of many the authors of this study having associations with pharma are listed. Given the very high degree of consensus observed here, a potential limitation could be that given that many authors have associations with relevant pharma companies, this could have been a factor which played into the statement development itself (unconsciously). This should be added clearly to the limitations.

Agree, this has been added as a limitation.

4. The CKD statements themselves are not surprising and represent an optimal approach to management. The value here is the collations of the statements together, but most of the statements are not novel.

Agree

There are some areas where more emphasis would improve the relevance of this manuscript:

5. Many of the suggestions are quite high-resource setting focused. Some discussion of the feasibility of implementation of the recommendations across the country income spectrum and prohibitive medications costs/unavailability should be included in the discussion, for example referencing the Global Kidney Health Atlas report on lack of availability of basic diagnostics at primary care in low resource settings. MDTs are certainly not available world-wide, please include comments about this as well.

Thank you for this feedback, we agree that these issues are not adequately mentioned, we have added more detail and included the reference you suggested to the 1st section of the discussion

6. There is no discussion of coverage for CKD care - the sustainability of the suggested strategies should be discussed, especially thinking about lower resource settings. How should the best be covered financially, integrated into the health system? Quality monitored/assured?

We have updated the discussion of **Holistic management of CKD in cardio-renal-metabolic patients** to include some further considerations around the management of CKD in lower resource settings, including the emphasis on earlier detection and management to reduce progression to more costly disease states (e.g., dialysis), and how savings made through this initiative could be reinvested to further improve care.

7. More emphasis should be placed on earlier identification more consideration will need to be given to identifying patients with “non-traditional” risk factors, especially in lower resource settings, should be highlighted more strongly. Who is high risk where?

In addition to above, we have added the need for awareness of communicable risk factors in lower resource settings (particularly with reference to Africa). These discussion also highlights the need for investment in basic primary care and diagnostic services in LMICs.

8. The methods needs to be clearer, how were respondents identified/reached? Could this have led to some bias?

We have added some detail regarding the gathering of responses via a third-party (Sermo), this was intended both for convenience but also to avoid any selection bias (for example, if we had used a snowball method)

9. Why is COVID specifically included in the AKI statement 14?

This was specifically added due to relevance at the time of the steering group discussion with emerging reports of the link between COVID-19 and AKI.

10. Patient empowerment, health literacy, possibly even patient navigators were not represented adequately in the statements/discussion, this should be included as a limitation

This has been added as a limitation as requested

11. A final call for affordability and availability of the recommended medications (i.e. reducing prices) everywhere should be made, despite the study being funded by pharma.

This has been added in the discussion of **Cross-specialty alignment** as requested.

12. Lines 27/28 on page 8 seem incomplete sentences?

Apologies, this has been corrected,

Reviewer: 2

Comments to the Author:

1. There is a need to define the KDIGO acronym in full on page 7 (line 11) as I believe this is the first time it appears in the main manuscript.

Apologies, this has been corrected,

2. The authors should consider reporting the study according to established guidelines - e.g. CREDES <https://www.equator-network.org/reporting-guidelines/credes/> - which should be highlighted at the start of the methods section.

Thank you for this suggestion, we have added this in along with a CREDES checklist as supporting information.

3. For completeness (and transparency) ideally, there should be more demographic details regarding the steering group membership and any conflict of interests.

We have added more detail regarding Steering Group demographics and conflicts of interest have been listed in full in the declarations

4. Presenting the theme titles in isolation on pg.8 somewhat hinders interpretation. Some further explanation of each theme is required to give context for later sections.

Agree, we have removed the themes from page 8 and presented them in the results

5. It is a little unclear how the steering group rating of statements, led to the final statement set used for the initial round of consensus. For example, what proportion of 'accept' ratings was required to include a particular statement in the final statement set?

We have updated the methods to describe how the statements were ratified by the steering group based on simple majority with the potential for further group consultation for any significant differences of opinion on the fundamental principles of any statement.

6. The stopping criteria should be justified. Were the criteria selected according to an evidence-based approach?

Yes, we have added a reference to Diamond et al and expanded on the stopping criteria to provide transparency and improve reproducibility.

7. When determining whether a particular statement met the consensus agreement threshold, presumably both the 'tend to agree' and 'strongly agree' responses were aggregated into a single agreement metric? This approach should be outlined in the methods. I note that the authors state 'surveys were collated... in line with Delphi methodology...', but I feel more information is needed here.

Agree, we have explicitly added this information to the method as requested.

8. I feel that many might take issue with the following statement:

"Patients and the public were not involved in this research as it is primarily concerned with healthcare policy and practice."

Reasoning that patients' opinions and preferences should play a more important role in healthcare policy and practice. With regards the current study, however, arguably the justification for the non-inclusion of patients and the public should be linked more to the stated research objective on pg.6.

Agree, we have aligned the reason with the objective as suggested, and added the lack of patient experience as a limitation.

9. I feel it is important to breakdown the agreement data in Table 1. It should be clear for each statement what proportion of respondents selected the 'strongly agree' versus the 'tend to agree' option. This allows the reader to explore the differing nature of responses for statements which scored a similar overall level of agreement. For example, both item 9 and item 24 scored similarly for their overall agreement, but the proportions of those respondents strongly agreeing with each statement was quite different (Item 24 = SA 27%, TTA 65%; Item 9 = SA 48%, TTA 44%).

Agree, this is good to know for future work and we have broken this down as requested in Table 1.

10. In the discussion, some statements required slightly more explanation/qualification as they appeared to extend beyond the scope of the study findings. For example:

“Earlier identification & screening of CKD (S1-9)

There was very high agreement (>90%) with all statements in this topic, and the key principles that early diagnosis of CKD is key to implementing strategies to slow disease progression. To this end, national kidney health screening and diagnostic programmes should be implemented.”

It is the last statement here that is somewhat problematic. I feel it might be more accurate to state that the views of the Delphi respondents suggested there was strong consensus supporting the need for national kidney health screening and diagnostic programmes. Adjusting the language in this way serves to keep the recommendations grounded in the study data/findings. I note that this was done at times during the discussion - e.g. pg.15 ‘Very high levels of agreement suggest strong support for multidisciplinary involvement in CKD management...’ – but, perhaps needed to be a little more consistent throughout.

Thank you, we have amended as suggested and tried to adjust the language of the discussion to ensure consistency in this approach.

11. I note the strengths and (one) limitation were presented in bullet point format rather than discussed. The CREDES Delphi reporting guidelines suggest:

“Discussion of limitations. Reporting should include a critical reflection of potential limitations and their impact on the resulting guidance.”

The authors should consider addressing this in the final manuscript.

Agree, we have corrected this and expanded the limitations in line with feedback received.

VERSION 2 – REVIEW

REVIEWER	Luyckx, Valerie Institute of Biomedical Ethics and History of Medicine
----------	---

REVIEW RETURNED	12-Jan-2024
-------------

GENERAL COMMENTS	I may be missing it but I could not find a point-by-point response to the reviewers comments. From the tracked changes I can see that changes have been made. There are however remaining concerns.  1. The authors state that only 11 countries were targeted. Please explain why these 11 countries were selected, i.e. why representatives from these countries were included in the Steering group and not others? The authors have not adequately addressed the fact that more low resource representation would have been crucial here. 2. In the added section of the methods on page 37, line 3 the authors state “outcomes for CKD ...are particularly poor in LMICs, to understand barriers and opportunities, a modified Delphi approach was employed...” This statement leads the reader to understand that the authors planned to use this study to address problems in LMICs, but in fact without including experts form LMICs (beyond Egypt, where access to care is better than in many other LMICs)? This is problematic if this was indeed the case? 3. Many comments have been added about low resource settings but these comments do not address the lack of local expert input here. Some references relating to this are outdated, e.g. reference 9. The most quoted study published in 2015 states 2 – 7 million people with kidney failure may die without dialysis. Please update the references. 4. Similarly the highlighted section added on page 41 discusses LMICs, but the entire issue of screening in LMICs is far from resolved, who to screen, when to screen, how to screen, who will pay? These issues again could really only be contributed by local experts. Proposing investment in LMICs is not necessarily wrong, but is highly simplistic. 5. I would suggest this paper be framed from the perspective of the 11 countries, sticking to the conclusion reached by the existing experts (which I do not dispute are useful and valid and that the exercise per se has value). The major limitation here is that LMIC representation was suboptimal and therefore extrapolation of the conclusions to LMICs would require a further Delphi process involving local experts. This is mentioned in 1 line on page 46, but this is a crucial issue that the reader must realize. The findings here are not necessarily generalizable to LMICs. 6. Thank you for clarifying that the authors received honoraria from Astra Zeneca. This is an important declaration for transparency.
--

REVIEWER	Kyte, Derek University of Worcester
REVIEW RETURNED	26-Jan-2024

GENERAL COMMENTS	I thank the authors for addressing all previous comments.
---

VERSION 2 – AUTHOR RESPONSE

1. *The authors state that only 11 countries were targeted. Please explain why these 11 countries were selected, i.e. why representatives from these countries were included in the Steering group and not others? The authors have not adequately addressed the fact that more low resource representation would have been crucial here.*

Thank you for your feedback. We have added further detail to the selection process which is hopefully addresses this comment, we have also added to the strengths and limitation section regarding the lack of LMIC representation (See point 4) :

'The steering group members were selected to provide representation across a range of development indicator levels external to Europe and North America (to avoid replication of previous published work). Steering Group members were recruited to represent countries outside of North America and Europe. Central and South America, South East Asia, Middle East and Africa were initially targeted. The process of recruitment involved identification of potential group members from each of these regions using desk research, followed by a snowball method until all target regions had at least one representative on the Steering Group.'

2. *In the added section of the methods on page 37, line 3 the authors state “outcomes for CKD ... are particularly poor in LMICs, to understand barriers and opportunities, a modified Delphi approach was employed...” This statement leads the reader to understand that the authors planned to use this study to address problems in LMICs, but in fact without including experts form LMICs (beyond Egypt, where access to care is better than in many other LMICs)? This is problematic if this was indeed the case?*

We have removed this statement in line with Point 4 below. Hopefully this removes concerns regarding the focus and remit of the paper.

3. *Many comments have been added about low resource settings but these comments do not address the lack of local expert input here. Some references relating to this are outdated, e.g. reference 9. The most quoted study published in 2015 states 2 – 7 million people with kidney failure may die without dialysis. Please update the references.*

The changes related to Point 4 have removed these references, they have also removed the discussion around LMICs as this is not within scope of the paper.

4. *Similarly the highlighted section added on page 41 discusses LMICs, but the entire issue of screening in LMICs is far from resolved, who to screen, when to screen, how to screen, who will*

pay? These issues again could really only be contributed by local experts. Proposing investment in LMICs is not necessarily wrong, but is highly simplistic.

We agree that discussion of the issues concerning LMICs goes beyond the scope of this paper given the countries involved and have removed the references to LMICs from the text to avoid any inference to the contrary.

- 5. I would suggest this paper be framed from the perspective of the 11 countries, sticking to the conclusion reached by the existing experts (which I do not dispute are useful and valid and that the exercise per se has value). The major limitation here is that LMIC representation was suboptimal and therefore extrapolation of the conclusions to LMICs would require a further Delphi process involving local experts. This is mentioned in 1 line on page 46, but this is a crucial issue that the reader must realize. The findings here are not necessarily generalizable to LMICs.*

We have reframed the paper is directed by removing any aspects of the introduction, methods and discussion that suggest that this paper is targeted at LMICs or that the findings are directly applicable to LMICs. We have also added wording in line with your suggestion above to the discussion of Strengths & Limitations on Page 14 of the paper...

'The lack of representation on the Steering Group (and subsequently the wider panel) of members from LMICs was a significant limitation. Whilst the survey covered 11 countries of varying economic status, the range was from lower-middle income to high, a clear gap exists, namely those countries classified as 'low income' (e.g., Sub-Saharan Africa). Consequently, the findings of this study may not be generalisable to LMICs. In order to make such recommendations a further Delphi process would be required specifically to engage with this demographic, this would provide insight into the specific challenges and considerations and allow for comparison with the current dataset.'

...and also to the first line of the conclusion:

'The Steering Group was able to form a set of recommendations specific to the 11 participant countries ranging from lower-middle income to high income and relevant to other countries with a similar demographic.'